Demography and homing behavior in the poorly-known Philippine flat-headed frog Barbourula busuangensis (Anura: Bombinatoridae)

Miñarro Marta 1 marmiro@mncn.csic.es
Burrowes Patricia 1 2
Lansac Claudia 1
http://orcid.org/0000-0001-8100-1061 Sánchez-Montes Gregorio 1
Afuang Leticia E. 3
De la Riva Ignacio 1 iriva@mncn.csic.es
1 Departamento de Biodiversidad y Biología Evolutiva, Museo Nacional de Ciencias Naturales (MNCN-CSIC) , Madrid , Spain
2 Department of Biology, University of Puerto Rico , San Juan, Puerto Rico , United States
3 Animal Biology Division, University of the Philippines , College Laguna, Los Baños , Philippines
Manjarrez Javier
Electronic publication date: 2025 Jan 14
Publication date: 2025
Volume: 13
Electronic Location ID: e18694
Received 2024 Oct 1; Accepted 2024 Nov 20
Copyright: © 2025 Miñarro et al.
Copyright year: 2025
Copyright holder: Miñarro et al.
License: This is an open access article distributed under the terms of the Creative Commons Attribution License, which permits unrestricted use, distribution, reproduction and adaptation in any medium and for any purpose provided that it is properly attributed. For attribution, the original author(s), title, publication source (PeerJ) and either DOI or URL of the article must be cited.
License URL: https://creativecommons.org/licenses/by/4.0/

Keywords: Amphibia, Abundance, Age structure, Capture-mark-recapture, Site fidelity

Funding: Palawan and Busuanga CGL2014-56160-P, PGC2018-097421-B-I00, MCIN/AEI/10.13039/501100011033 European Union Patricia Burrowes NSF-IOS-2011281 Spanish Ministry of Science and Innovation The research of Ignacio De la Riva, Marta Miñarro, and Patricia Burrowes in Palawan and Busuanga was financed by projects CGL2014-56160-P and PGC2018-097421-B-I00 (PI: I. De la Riva), funded by MCIN/AEI/10.13039/501100011033 and by “ERDF A way of making Europe”, by the “European Union”. Research by Patricia Burrowes was supported by NSF-IOS-2011281. Marta Miñarro was supported by a Ph.D. fellowship of the Spanish Ministry of Science and Innovation. There was no additional external funding received for this study. The funders had no role in study design, data collection and analysis, decision to publish, or preparation of the manuscript.

==============================
The flat-headed frog, Barbourula busuangensis, is a poorly known, riverine species, endemic to the province of Palawan in the Philippines. We applied capture-mark-recapture (CMR) methods to follow individuals at two sites (Malbato and San Rafael) in the island of Busuanga over 10 months in 2022–2023. We used passive internal transponders (PITs) to mark adult and subadults and single-colored visual internal elastomers (VIEs) for cohorts of juveniles. From a total of 196 frogs PIT-tagged in Malbato and 144 in San Rafael, we obtained overall recapture rates of 49% and 60% respectively. We used the POPAN formulation in MARK software to estimate abundance, survival, movement, and age-class demographics. Our best model estimated an average population size of 268 frogs at Malbato and 232 at San Rafael, and constant survival probabilities (mean ≥ 0.97) at both sites. When adding age classes to the model, abundance of adults was always higher than that of subadults producing an age structure dominated by adults at both sites. Growth rates decreased significantly with body size, being higher in juveniles (1.51 mm/month) and subadults (1.56 mm/month) than in adult frogs (0.60 mm/month). At these growth rates frogs may reach adulthood at 2.5 years, with the oldest individuals likely being over 11 years old. CMR data confirmed site fidelity, and translocation experiments revealed that frogs have the ability to home when displaced 10–50 m upstream and downstream from their original capture site. This is the first long-term study of B. busuangensis using robust field and analysis methods. Our data suggest that B. busuangensis is stable at present in Busuanga, with long-lived adults and dispersing subadults. We expect that these data may serve as baseline of current population abundance, age structure and growth rates which are factors that tend to be affected when species are threatened. In this way it may help researchers and conservation practitioners detect potential changes that may occur as this species confronts the challenges of the Anthropocene.

Introduction

The Amphibia are the most endangered class of vertebrates, with 40.7% of species catalogued as threatened by the International Union for the Conservation of Nature (IUCN) (Luedtke et al., 2023). Among many factors, habitat degradation, climate change, and emerging infectious diseases are considered the most important drivers of global amphibian declines (Cohen et al., 2019; Luedtke et al., 2023). Chytridiomycosis, a disease caused by the pathogenic fungus Batrachochytrium dendrobatidis (Berger et al., 1998; Longcore, Pessier & Nichols, 1999) is of main concern because it has been associated with the extinction or decline of many amphibian species worldwide (Scheele et al., 2019). However, in Southeast Asia, a hotspot for biodiversity and endemism, amphibians are mostly threatened by habitat loss due to deforestation (Rowley et al., 2010; Turner & Snaddon, 2023). Although the data for deforestation and habitat degradation in Southeast Asia are not complete or equally documented per country, it has been estimated to have lost more than half of its original forest cover (Turner & Snaddon, 2023; Chen et al., 2023). The major land use changes leading to deforestation in this area are the conversion of forests to agriculture, logging, and drought (Chen et al., 2023). Illegal gold mining has also contributed to deforestation and, additionally, it results in habitat contamination of water sheds affecting both habitat and biodiversity (Bickford, Iskandar & Barlian, 2008). For example, the Bornean endemic anuran Barbourula kalimantanensis Iskandar, 1978, considered Endangered (EN) by the IUCN SSC Amphibian Specialist Group (2018), has been affected by these practices and may already be on the brink of extinction (Bickford, Iskandar & Barlian, 2008).

According to the Global Amphibian Assessment (2006), 81% (235/348) of the amphibians in Southeast Asia are endemic to this region (Pratihar et al., 2014). Despite the high number of endemic species in southeast Asia, amphibians from this region have remained relatively unnoticed by the global conservation community most likely owing to the lack of knowledge about their distribution, population status, or natural history (Rowley et al., 2010). As a result, there is insufficient data for reliable conservation status assessments, leading researchers to suggest that the number of species catalogued as threatened may represent an underestimation (Hoffmann et al., 2010; Alcala et al., 2012; Pratihar et al., 2014). This may be grave because without a clear knowledge of how many and which species are threatened, it is difficult to make conservation prioritization leading to effective management strategies (Nowakowski et al., 2024). Thus, there is a need for studies on the demography, abundance, and natural history of amphibians in Southeast Asia in order to put forward evidence-based research questions and conservation.

Natural history studies are especially valuable in the case of poorly documented species, because they may expose unknown behaviors and ecological strategies over which further questions can be raised or management practices applied (Bury, 2006). For example, work on the native Puerto Rican frog, Eleutherodactylus coqui, Thomas, 1966 unexpectedly revealed that this species has internal fertilization (Townsend et al., 1981) and male parental care of direct-developing embryos (Townsend, Stewart & Pough, 1984). These findings promoted further research on evolutionary developmental biology, ecology and physiology (reviewed by Joglar (1998)), and now this species is considered a model organism (Westrick, Laslo & Fischer, 2022). In Venezuela, studies on the critically endangered frog Atelopus cruciger (Lichtenstein & von Martens, 1856) found that it exhibits strong site fidelity when breeding. This behavior was pivotal for establishing conservation strategies, including captive breeding programs with future reintroductions (Señaris et al., 2023).

Together with natural history research, studies on the distribution, abundance and population dynamics of amphibians are key to determine their conservation status (Miller et al., 2007). Demographic parameters like population size, survival, recruitment and dispersal rates are fundamental to characterize population fluctuations and assess extinction risk in the face of current and future threats (Newell, Goldingay & Brooks, 2013; Fernández de Larrea et al., 2021; Señaris et al., 2023). For these purposes, capture-mark-recapture (CMR, hereafter) is one of the most reliable techniques (Lebreton et al., 1992; Clutton-Brock & Sheldon, 2010), and can also inform on habitat use spatial behaviors such as site fidelity and homing (Balázs, Lewarne & Herczeg, 2020; Capellà-Marzo, Sánchez-Montes & Martínez-Solano, 2020).

The Philippines archipelago, located within the southeast Asian biodiversity hotspot, is inhabited by at least 106 species of amphibians (Frost, 2024), from which the latest IUCN assessment catalogued 26 of them under different categories of threat (Vulnerable, Endangered and Critically Endangered, IUCN, 2024). Since there is almost no demographic information on Philippine amphibians, the list of endangered species is likely underestimated. Nevertheless, over 90% of amphibians in the Philippines are considered at risk due to projected climate change evidenced by warming temperatures, increasing intensity of dry periods, and greater frequency of extreme weather events like typhoons (Alcala et al., 2012). As a further concern, many Philippine amphibians have limited geographical ranges and/or occur in isolated habitat fragments where disturbances could potentially lead to significant threats (Diesmos et al., 2002). Thus, the Philippines represents a critical area of southeast Asia to study amphibians before all the above-mentioned threats may aggravate their situation.

In this article, we address demographics and behavioral aspects of a poorly studied Philippine frog, Barbourula busuangensis Taylor & Noble, 1924 (Figs. 1A–1B). This species is endemic to the island of Palawan and some adjacent islands, like Busuanga in the north, and Balabac in the south (Fidenci, 2007; Frost, 2024). Due to its limited distribution and potential anthropogenic threat, it is considered Near Threatened (NT) by the IUCN SSC Amphibian Specialist Group (2018), but the lack of demographic data has prevented an accurate conservation status assessment. Barbourula busuangensis is a fully aquatic, nocturnal frog that inhabits clear, rocky, well oxygenated forest streams in tropical rainforests (Diesmos et al., 2015). It is surprising that such an interesting frog, described a century ago, with only one other species in its genus, and being within a primitive clade (Bombinatoridae), is so poorly known. For example, its advertisement call was described just recently (Bosch et al., 2023), and there is very limited information about its distribution, ecology and natural history (Flores et al., 2024). In contrast, thorough studies on the anatomy of its muscular-skeletal system (Clarke, 1987; Přikryl et al., 2009; Roček et al., 2016) and on its phylogenetic relationships (Blackburn et al., 2010) are available. To leverage this gap of information, we conducted extensive field work to study B. busuangensis in its native environment. We used CMR methods and multistate modeling to estimate population size, age structure, growth rates and explore the potential for homing behavior. Due to the lack of ecological data for this species, specific hypotheses were challenging to formulate. Thus, this represents a descriptive study that offers for the first-time robust information about demography of B. busuangensis and provides baseline data for the assessment of the population status of this enigmatic and potentially threatened species. We expect that further research will lead to hypothesis-driven questions that will advance knowledge from an ecological, evolutionary and conservation point of view.

Figure 1 Adult and juvenile of Barbourula busuangensis and its typical habitat at Malbato.

Barbourula busuangensis (A) adult, and (B) juvenile, photographed at our field site of Malbato (Busuanga, Palawan, Philippines) by Javier Aznar; (C) typical habitat of B. busuangensis at Malbato, photo by MM.

Materials and Methods

Study area and CMR monitoring

This study was conducted in the island of Busuanga, in the Calamian Archipelago, province of Palawan, Philippines (Figs. 2A–2B). Busuanga has an area of 915.16 km2 and a maximum elevation of 620 m a.s.l. It is known for its rich and diverse vegetation, including, in order of abundance, early secondary growth forests, advanced secondary growth forest and old growth forests, land used for agricultural purposes, and mangroves (Supsup et al., 2023). The University of the Philippines at Los Baños and the Palawan Council for Sustainable Development approved the permit to conduct fieldwork in Busuanga (Wildlife Gratuitous Permit N° 2022-11).

Figure 2 Map depicting the study sites.

(A) Map of the Philippines; the star marks the location of Manila, and the arrow points at the island of Busuanga, (B) the island of Busuanga enlarged, showing our two study sites: Malbato in the south (orange) and San Rafael in the north (purple) and, (C) satellite images from Google Maps (Map data © 2024 Google) of the study sites with transects along rivers (blue lines) highlighted in orange (transects “A”, “B” and “C”), and purple.

We conducted field work over a total of 10 months, divided in three periods during 2022 and 2023, corresponding to April–July 2022, October–December 2022, and April–June 2023. We worked at two locations along rocky rivers surrounded by dense tropical rainforest (Fig. 1C). Our main study site, Malbato (Barangay Bintuan, 12°2′14″N, 120°5′49″E) (Fig. 2B, orange circle) is in the southern part of the island, likely near the vaguely described type locality of the species (Taylor & Noble, 1924). Here we established three transects adding up to 420 m (transect “A” = 100 m, transect “B” = 100 m and transect “C” = 220 m, Fig. 2C). Transects were marked with flagging tape every 10 m to identify specific locations where frogs were captured. Due to logistical reasons, we sampled in Malbato more frequently for a total of 40 nights throughout our study period, and on average surveys were conducted by two or three people. The second site, San Rafael, is in the north-western part of Busuanga, in the municipality of San Rafael (Barangay New Busuanga, 12°12′12″N, 119°55′9″E) (Fig. 2B). At this site, we monitored Barbourula busuangensis along a 320 m transect once a month during each field period, for a total of nine nights (Fig. 2C). Field parties at this site were always composed of four people such that in a way, the man-effort at both localities was balanced. The mean temperature during our work in Busuanga was 28 °C (range 26–31 °C) and the mean daily precipitation varied from 4.86–7.13 mm/day (data obtained from visualcrossing.com).

We monitored our transects along the river from 6:00 PM to approximately 3:00 AM, aiming to detect and capture every Barbourula busuangensis in the study area. For every captured individual, we measured body size as the snout-to-vent length (SVL) to the nearest 0.01 mm using a manual caliper. Individuals measuring more than 35 mm were marked with an 8 mm mini-Passive Internal Transponder (PIT) (ID 100A/1.4; Trovan, Douglas, Isle of Man, UK). To insert the transponder, we lifted the skin of the upper dorsal area of the frog with fine-tip tweezers and used surgical scissors to make a small incision. We placed the loaded canula (NDL-1xx/1.4; Trovan, Douglas, Isle of Man, UK) in the pistol grip implanter (IM-300C; Trovan, Douglas, Isle of Man, UK) at the incision point and inserted the microchip under the skin. Then, we used the blunt end of the tweezers to push the transponder posteriorly, close to the urostyle area. We did not seal the wound after insertion because experience with other tropical frogs proved that using this method yielded high transponder retention and no signs of infection (Burrowes, unpublished data). Each transponder has a unique digital code that can be detected with a portable reader (LID-573; Trovan, Douglas, Isle of Man, UK). Frogs with SVL < 35 mm were considered too small to be tagged with transponders and were marked with visual internal elastomers (VIE) instead. VIEs consist of a UV-fluorescent polymer that is injected subcutaneously using a 29–gauge syringe (Northwest Marine Technology Inc., Anacortes, WA, USA).

Age structure

To assess the age structure of the populations at both study sites, we classified individuals in six categories based on their SVL: Juveniles-I (individuals with SVL < 25 mm), Juveniles-II (SVL = 25–34 mm), Subadults (SVL = 35–49 mm), Adults-70 (SVL = 50–70 mm), Adults-90 (SVL = 71–90 mm) and Adults-100 (SVL > 90 mm). Because Barbourula busuangensis lacks obvious morphological features signaling sexual maturity, our criterion for determining adulthood was informed by the size of the smallest gravid female observed during our study (SVL = 51.2 mm); then, the younger age classes were subjectively estimated. Although skeletochronology (see for example Castanet & Smirina, 1990; Roček et al., 2016; Rahman et al., 2022) or dissection to observe gonad development (see for example López et al., 2017; Piazza et al., 2023) are more accurate methods to assess age and adulthood respectively, they require sacrificing individuals. Our work was limited in this respect, because we did not have permits to collect animals for the preservation. We applied different VIE colors (coded by fieldwork period, Table 1) to distinguish younger juveniles (Juveniles-I) from larger, and presumably older ones (Juveniles-II). These different color codes allowed us to recognize and track young cohorts through their growth until they reached subadult age class, at which we marked them with PITs (Peña-Jiménez & Burrowes, 2021).

Table 1 Number of Barbourula busuangensis at age class Juveniles-I (SVL < 25 mm) or Juveniles-II (SVL = 25–34 mm) marked with specific VIE colors and recaptured during the different field seasons of our study.

When VIE-marked juveniles were recaptured in subsequent field seasons, they were either re-marked with a new VIE color corresponding to their age class for that field season (e.g., (Blue) + Pink) or tagged with a transponder ((Blue) + PIT-tag) if they grew to subadults (SVL ≥ 35 mm).

			MALBATO	SAN RAFAEL	
Field season	Age class	VIE color	Marked	Recaptured	Marked	Recaptured	
Jun–Jul 2022	Juveniles-I	Yellow	6	1	1	1	
Juveniles-II	Blue	8	5	18	1	
Oct–Dec 2022	Juveniles-I	Green	7	2	9	0	
Juveniles-II	Pink	23	13	3	0	
Juveniles-II	(Blue) + Pink	6	6	0	0	
Subadults	(Blue) + PIT-tag	2	2	0	0	
Apr–Jun 2023	Juveniles-I	Orange	16	3	4	0	
Juveniles-II	Purple	8	3	1	0	
Juveniles-II	(Pink) + Purple	10	10	0	0	

Since Barbourula busuangensis also lacks external sexual dimorphism, we could only confirm the sex of gravid females when mature oocytes were visible through the ventral skin (Bosch et al., 2023). Consequently, we could not compare results by sex or determine sex biases in our sampling. To assess if the age structure was different between the southern (Malbato) and northern (San Rafael) populations, we conducted a Chi-square test of independence by sampling site.

Growth rates and age estimates

We estimated growth rates of post-metamorphic frogs as the increase in body size (SVL) divided by the time lapse between captures (mm/month) for both cohorts of juveniles, and for adult and subadult frogs marked with transponders. We applied a generalized lineal model (GLM) to explore the relationship between the SVL of adult and subadult frogs at first capture (explanatory variable), and their growth rate per month at subsequent recaptures (response variable). Because the response variable (monthly growth rate) adjusted to a gamma distribution, we implemented the family = gamma (link = “log”) arguments within the glm function in “R” statistical package to run the model, and qqplots and moments package to check the normality of residuals (R Core Team, 2020). We used average growth rates by age-class category to extrapolate the approximate age (in years) of Barbourula busuangensis of specific body sizes.

Population size estimates

We used MARK software (White & Burnham, 1999) to analyze the capture histories of all frogs marked during the study period. Specifically, we applied the POPAN formulation (Schwarz & Arnason, 1996) to estimate population size. This formulation is appropriate for the analysis of open populations where births, deaths, and movement of individuals in and out of the study area are likely to occur (Capellà-Marzo, Sánchez-Montes & Martínez-Solano, 2020; Fernández de Larrea et al., 2021). The POPAN models include four main parameters: (1) apparent survival of individuals (ϕ), (2) probability of capture (p), (3) rate of entrance of new individuals into the study area (pent), and (4) total population size, accounting for all the individuals present at the survey area during the study period (N). We tested a set of a priori biologically plausible models, with the parameters ϕ, p and pent as either constant (.) or time dependent (t) for both study sites. We analyzed the CMR datasets in two ways; first, to compare overall estimates between the north and south populations of Barbourula busuangensis, we analyzed each population separately by grouping individuals of all age classes over the three sampling periods at each site. Secondly, and only for Malbato, where the higher frequency of sampling occasions allowed for finer scale analysis, we divided individual capture histories by the three-field sampling periods, and included age-classes defined by body size as adults (>50 mm SVL) and sub-adults (35–49 mm SVL). In this detailed analysis at Malbato, ϕ, p and pent were modelled as either constant (.), dependent on age-class group (g), time (t) or their interaction (g*t). The likelihood of the tested models was assessed by Akaike’s information criterion corrected for small sample sizes (AICc), and estimates of population size (N) were obtained for each site based on weighted averaging among the AICc-ranked models. We used the software U-CARE to validate model assumptions via goodness of fit (GOF) tests (Choquet et al., 2009). The two most common sources of assumption violations are known as transience and trap-dependence effects, and both were examined using the tests 3.SR and 2.CT in UCARE, respectively. Transience effects caused by dispersing individuals violate the assumption that all frogs in the population have the same probability of being captured, while trap-dependence considers that marking has some effect (either positive or negative) on the probability of subsequently recapturing an individual (Pradel & Sanz-Aguilar, 2012).

Potential for homing behavior

We conducted displacement experiments at Malbato to test for the potential of homing behavior, defined as the probability of an individual to return to its original capture site after being displaced. Only marked frogs that had been recaptured at least once at the same location, were regarded as residents and considered for displacement experiments (Crump, 1986). We first moved 32 individuals 10 m away from their capture site; specifically, we translocated 10 adults and six subadults 10 m upstream, and nine adults and seven subadults 10 m downstream. We considered that an individual had exhibited homing behavior when recaptured within a 5 m radius of its original capture site. Once we had evidence that some individuals were returning to the original capture site, we extended the displacement distance to 30 m and 50 m during our second and third sampling periods. A total of 54 adults and 28 subadults were displaced at increasing distances (10, 30 and 50 m) upstream and downstream. During displacements, we collected frogs in transparent bags and carried them within a larger black bag. We did not run a control experiment to test if frog manipulation (i.e., capture and bagging) influenced the probability of homing. We used a multiple logistic regression model in R (R Core Team, 2020) to assess the effect of age class (adult or subadult), direction of displacement (upstream or downstream), and displacement distance (10, 30 or 50 m) on the probability of homing.

Permits to conduct this research in the field were obtained from Palawan Council for Sustainable Development (PCSD, Republic of the Philippines) and Ethics Competence B&C from Consejería de Medio Ambiente y Ordenación del Territorio de la Comunidad de Madrid, Spain. Ref. 10/096442.9/13.

Results

Capture-mark-recapture histories

We registered a total of 1,251 capture events at Malbato, including all age classes. At this site, we captured and marked with transponders a total of 196 individuals of Barbourula busuangensis corresponding to adult (143) and subadult (53) age classes (SVL > 35 mm). Among them, 73 adults and 23 subadults (49%) were recaptured only once, 42 adults and seven subadults were recaptured twice, 13 adults and five subadults were recaptured three times, five adults and six subadults were recaptured four times, and 10 adults and 12 subadults were recaptured more than five times, including one subadult that was recaptured 14 times (Fig. 3A). At San Rafael, we recorded a total of 1,042 capture events across all age classes and marked 144 individuals with transponders corresponding to 112 adults and 32 subadults. Of these, 74 adults and 15 subadults (60%) were recaptured only once, 25 adults and nine subadults were recaptured twice, seven adults and six subadults were recaptured three times, and six adults and one subadult were recaptured four times, with a maximum of seven recaptures for a single subadult (Fig. 3B).

Figure 3 Recaptured Barbourula busuangensis.

Top: Number of adult (black) and subadult (light grey) individuals of Barbourula busuangensis that were captured once and recaptured subsequent times at each of our study sites, (A) Malbato and (B) San Rafael. Bottom: Cumulative total number of frogs (adults and subadults) captured and PIT-tagged (dark grey) or recaptured (white) through time at (C) Malbato and (D) San Rafael.

We captured more adults than subadults at both sites (Figs. 3A–3B), and adults comprised 73% and 78% of the recaptured individuals in Malbato and San Rafael, respectively. The total number of animals captured and recaptured increased with our sampling effort and recapture rates per sampling month (total number of individuals recaptured over total marked) approximated 50% of the population at both sites by the end of our study (Figs. 3C–3D). Thus, marking with transponders proved to be a safe way to track Barbourula busuangensis in its riverine habitat. We did not observe any signs of losing the transponders contrary to findings in frogs of the genus Litoria (Brannelly, Berger & Skerratt, 2014), or infection at the incision point despite not sealing the wound. Moreover, recaptured frogs had sealed the wound between 2–4 days of initial marking.

VIE proved to be a safe marking method to follow young froglets, too small to withstand transponders. We captured and marked with VIE 68 juveniles (<35 mm SVL) in Malbato and recaptured 44 (65%) either in the same or in subsequent field seasons (Table 1). Eight of the juveniles-II marked with blue VIE during the first sampling season were recaptured during the second season; six of these maintained a body size of SVL < 35 mm, and thus, were classified again as juveniles-II and re-marked with a pink VIE, whereas the other two were PIT-tagged as subadults because they had grown to SVL ≥ 35 mm (Table 1). Ten juveniles-II marked with pink VIEs during the second field season were recaptured during the third season, still at this age class, and re-marked with a purple VIE. In San Rafael, we marked 36 juveniles and recaptured only two of them during the first field season (Table 1).

Age structure

The proportion of individuals at each age class was independent of study site (X2 (5, 450) = 0.61, p = 0.96), yielding a similar age class distribution for Malbato and San Rafael (Fig. 4). Adults of SVL between 50 and 90 mm were the dominant age-class at both sites. In contrast, we recorded very few adults larger than 90 mm in SVL (Adults-100) during our study, with only four in Malbato and one in San Rafael. We observed 23 individuals that we could accurately identify as adult females because they had visible eggs from the venter indicating that they were gravid. The size of these females ranged from SVL = 50–69 mm (N = 5), SVL = 70–79 mm (N = 18), and SVL = 80–90 mm (N = 4). Despite the greater sampling effort at Malbato, we found a similar number of subadults at both sites, 44 in Malbato and 33 in San Rafael (Fig. 4). We recorded almost the same number of Juvenile-II individuals for both sites (27 in Malbato and 28 in San Rafael), but almost twice as many Juvenile-I in Malbato than in San Rafael (Fig. 4).

Figure 4 Age distribution of Barbourula busuangensis in our two studied sites.

Age distribution of Barbourula busuangensis in two populations at (A) Malbato and (B) San Rafael, in Busuanga, Palawan, for each of the six proposed age-classes: Juveniles-I (Juv-I, SVL < 25 mm), Juveniles-II (Juv-II, SVL 25–34 mm), Subadults (Sub, SVL 35–49 mm), Adults-70 (Ad-70, SVL 50–70 mm), Adults-90 (Ad-90, SVL 71–90 mm) and Adults-100 (Ad-100, SVL > 90 mm).

Growth rates and age estimates

For cohorts of VIE-tagged juveniles that were recaptured in subsequent field seasons in Malbato we estimated average growth rates of 4.7 mm (range: 0.4–11 mm) in SVL over a period of 3–4 months (1.21–1.50 mm/month). Thus, if we consider the smallest froglet observed (SVL = 10.3 mm) a recent metamorph, we can infer that it takes a very young juvenile of Barbourula busuangensis approximately 17 months (roughly 1.5 years) to reach a SVL of 35.8 mm equivalent to our subadult age class (Table 2). For larger individuals marked with transponders, monthly growth rate decreased significantly with SVL (coefficient = −0.035, p < 0.0001) (Fig. 5, Appendix S1 for output table). Subadults grew at a mean growth rate of 1.56 mm/month (95% C.I [0.17–2.64]), while adults grew at a slower rate, mean of 0.60 mm/month (95% C.I. [0.04–1.92]) (Fig. 5).

Table 2 Estimated growth rates and extrapolated age of post metamorphic Barbourula busuangensis based on increases on body size (SVL, mm) of recaptured individuals.

	Young Juvenile	Subadult	Adult-70	Adult-90	Adult-100	
SVL at start of age class	10.3 mm	36 mm	54 mm	72 mm	96 mm	
Mean growth rate per month	1.50 mm	1.56 mm	0.60 mm	0.4 mm	—	
Growth per year	18 mm	18.72 mm	7.2 mm	4.8 mm	—	
Time in years to reach next age class	1.5 years	1 year	2.5 years	5 years	—	
Age of older individuals	1.5 years	2.5 years	5 years	10 years	>11 years	

Figure 5 Scatter plot with fitted regression line showing the relationship between body size at first capture (SVL in mm) and the growth rate per month (difference in SVL) for recaptured individuals of Barbourula busuangensis.

Adults (SVL > 50 mm) are represented in black, and subadults (SVL 35–49 mm) in grey. Triangles represent individuals from Malbato, and circles for San Rafael (GLM, gamma distribution, coefficient = −0.035, p < 0.0001).

Using the mean growth rates obtained per age class we extrapolated the age of metamorphosed individuals by body size (SVL). In this way, a young subadult of 36 mm and 1.5 years old (according to mean juvenile growth rates-see above), growing at an average rate of 1.56 mm/month, may take 12 months to grow to a SVL of 54 mm, thus, reaching Adult-70 category roughly at 2.5 years of age. Then, it would take this frog about 2.5 more years, growing at mean adult growth rate of 0.6 mm/month, to reach 72 mm in SVL (corresponding to a young Adult-90 category), and another 4–5 years to reach SVL in the range of 81–90 mm (advanced Adult-90), making the oldest individuals in the population over 11 years old (Table 2).

Population size estimates

Considering the data for all individuals over the entire study period, the best-ranked model at each sampling site was that where apparent survival (ϕ) was kept constant, and the probability of entrance of individuals (pent) was variable across time (Table 3). However, while the probability of capture (p) was kept constant in the best model for Malbato, it was variable across time for San Rafael (Table 3; Appendix S2 for all models). The model-averaged population size in Malbato (N = 268, 95% confidence interval CI [242–293]) was slightly higher than that for San Rafael (N = 232, 95% CI [191–273], Table 3). These estimates indicate that we captured 73% (196/268) and 62% (144/232) of the population at both sites respectively. The estimated survival rate was high throughout our study, with a mean daily survival rate of 0.99 for both sites (Appendix S3 and S4). The probability of capture of individuals across the sampling sessions was notably higher in San Rafael (0.33), compared to Malbato (0.09, Appendix S3 and S4). On the other hand, the rate of entrance of new individuals into the study area was low for both sites (Appendix S3 and S4).

Table 3 Best ranked models yielded by the POPAN analysis in Malbato and San Rafael over the entire study period, with the model-averaged estimates of abundance (N) of Barbourula busuangensis.

Coding of model parameters: ϕ = apparent survival, p = probability of capture, pent = rate of entrance of new individuals into the study area, (.) = constant parameters, (t) = timedependent parameters.

Model	AICc	AICc weight	Parameters	Deviance	N	
Malbato						
Φ(.), p(.), pent(t)	2,155.49	0.999	42	215	266 (246–296)	
				N (model av.)	268 (242–293)	
San Rafael						
Φ(.), p(t), pent(t)	511.79	0.754	18	−422.1	233 (200–286)	
Φ(.), p(t), pent(.)	514.12	0.235	11	−402.8	228 (199–271)	
				N (model av.)	232 (191–273)	

When age class (adults and subadults) was included in the model and the three sampling periods were considered separately for Malbato, the best-ranked models were those where apparent survival (ϕ) was kept constant, the probability of capture (p) was dependent on the age class, and probability of entrance of individuals (pent) was variable across time and age classes (Table 4; see also Appendix S5 for all models). In the three sampling periods, a lower-ranked model showed age class dependent survival (Table 4). Daily survival rate in Malbato was concordant across all models, showing values ≥0.97 for all age classes in all sampling periods (Appendix S6, S7 and S8). The probability of capture, thus detectability, varied with age class, being on average higher for subadults (0.28) compared to that of adults (0.09). Finally, the mean probability of entrance of new individuals to the population was low across all sampling seasons and age classes (<0.001–0.265), but was generally larger for subadults than adults (Appendix S6, S7 and S8).

Table 4 Best ranked models yielded by the POPAN analysis in Malbato during the three sampling seasons (April–July 2022, October–December 2022 and April–June 2023) with the model-averaged estimates of abundance (N) for Barbourula busuangensis by age class.

Coding of model parameters: ϕ = apparent survival, p = probability of capture, pent = rate of entrance of new individuals into the study area, (.) = parameters held constant, (t) = parameters time-dependent, (g) = parameters dependent on age class, and (g*t) their interaction.

Sampling period	Model	AICc	AICc weight	Parameters	Deviance	N (subadults)	N (adults)	
Apr–Jul 2022	Φ(.), p(g), pent(g*t)	585.38	0.955	35	−75.5	28 (24–44)	174 (127–256)	
	Φ(g), p(g), pent (g*t)	591.50	0.044	36	−72.8	26 (23–44)	180 (132–265)	
					N (model av.)	29 (20–37)	174 (111–237)	
Oct–Dec 2022	Φ(.), p(g), pent(g*t)	466.30	0.811	25	−120.1	38 (32–53)	184 (128–289)	
	Φ(g), p(g), pent (g*t)	469.22	0.188	26	−120.1	38 (32–53)	186 (128–294)	
					N (model av.)	38 (29–48)	185 (107–263)	
Apr–Jun 2023	Φ(.), p(g), pent(g*t)	541.98	0.799	29	−56.6	39 (32–57)	167 (121–250)	
	Φ(g), p(g), pent (g*t)	544.75	0.200	30	−57.0	38 (31–55)	166 (121–247)	
					N (model av.)	40 (28–51)	167 (105–229)	

The abundance of Barbourula busuangensis estimated by this model varied between sampling periods for both age classes, but the model averaged estimates fall within the confidence intervals of the next sampling period (Table 4). While the average estimated abundance of subadults increased with time from the first to the last field season (N = 29–40), that of adults was highest in October–December 2022 (N = 185).

The GOF (goodness of fit) tests of the POPAN datasets for all individuals over the entire study period showed evidence of transience and trap-dependence effects in Malbato (Appendix S9). However, these effects were not consistently maintained when the data were analyzed by age-classes and field seasons. In this case, we only detected transience effects for subadults in two of the sampling periods, whereas no trap-dependence effect was found for any group or season (Appendix S9). We did not find any departures from POPAN model assumptions at San Rafael (Appendix S9).

Potential for homing behavior

We recaptured 53 of the 82 marked individuals that were displaced upstream or downstream, at time intervals between four and 353 days after their translocation. A total of 42 of the translocated individuals (51%) eventually returned to their original capture site, suggesting the probability of homing behavior in this species (Table 5). Seven individuals (8%, four adults and three subadults) were found elsewhere along the river, at an average distance of 63 m (range: 10–140) from their original capture site. The remaining 29 individuals (31%) were not recovered by the end of the study. Results of the multiple logistic regression model (Model: Homing ~ Age + Distance + Direction) revealed that the probability of homing was not significantly associated to age class (adults vs. subadults), distance of displacement (10, 30 or 50 m) or the direction of river flow (upstream vs. downstream) (Appendix S10).

Table 5 Summary of results of displacement experiments to study potential homing behavior in Barbourula busuangensis along the river at the Malbato study site.

Distance displaced (m)	Direction	Number adults displaced (homed)	Homing rate (%)	Number subadults displaced (homed)	Homing rate (%)	
10	Upstream	10 (6)	60	6 (6)	100	
	Downstream	9 (4)	44	7 (5)	71	
30	Upstream	5 (1)	20	5 (5)	100	
	Downstream	6 (5)	83	4 (3)	75	
50	Upstream	13 (5)	38	2 (0)	0	
	Downstream	11 (1)	9	4 (1)	25	

Discussion

Using CMR methods we obtained high recapture rates of Barbourula busuangensis. Our results provide a robust estimation of overall population sizes with little variation in abundance between study sites (Malbato: mean N = 268, San Rafael: mean N = 232; Table 3), and similar age class demographics over three distinct field seasons at Malbato (Table 4). Thus, abundance of B. busuangensis appears to be stable at present, and the fact that these numbers are not drastically different from those found in other field assessments further supports this hypothesis. For example, in surveys carried out not far from San Rafael, L. E. Afuang & K. Cielo (2010, unpublished data) captured 229 individuals between November 2008–May 2009; similarly, Flores et al. (2024) observed 270 individuals in 7 days along two river systems in northern Busuanga.

Capture history informed multistate models revealed that survivorship of individuals (adults and subadults combined) was high (>99%) and remained constant at both study sites (Appendix S3 and S4). These results are comparable to those found for other bombinatorid frogs. For example, Bombina pachypus showed a 96.2% return rate (product of survival and capture probabilities) and high survivorship in nearly all the sampling sites in central Italy (Angelini et al., 2018). For Bombina variegata, an aquatic breeder studied in Switzerland, survivorship was also high but varied (0.85 and 0.71) among distinct sites depending on whether water levels were variable or stable during the breeding season (Cayuela et al., 2019). In other riverine species, like the torrent frogs of the genus Hylodes from Brazil, survivorship was variable; while H. asper had a survival rate of 98% (Guimarães, Doherty & Munguía-Steyer, 2014), it was 38% in H. heyeri, potentially due to the effect of heavy rains causing frogs to be carried away from the population (Struett, Roper & Moura, 2023).

Despite recording over 1,000 capture events of Barbourula busuangensis at both localities, results from our best models across the entire study period found low capture probabilities in Malbato and San Rafael that varied with time at the latter site (Appendix S3 and S4). Models including sampling period and age class for the population in Malbato revealed that the mean capture probability was higher for subadults than for adults (Appendix S6, S7 and S8). These results were not surprising because subadults were easier to catch than adults. While adults are active in rock crevices between large boulders in the middle of the river and hide when they sense danger, younger frogs are found exposed in the riverbank often sitting on top of rocks, an observation also made by Flores et al. (2024). Thus, there seems to be a spatial separation by age classes in the activity of B. busuangensis that may favor detectability of subadults. Higher capture probabilities at San Rafael (Appendix S4) are likely due to increased sampling effort per night. Although we worked in San Rafael less often, expeditions to this area included larger field teams, typically of four vs. two people.

Our models for Malbato detected transience (migration) effects only for subadults during the first and last sampling periods, coincidental with the rainy season. In addition, the best models showed that both probabilities of capture (p) and migration (pent) were influenced by age class at this site during all our field seasons (Appendix S5). This finding suggests that subadults may be more prone to disperse than adults, perhaps to establish territories, and that migration may be higher during the rainy seasons.

Being able to age individuals by body size is interesting from a natural history perspective because we can assess age structure and make inferences about recruitment and longevity. It is also useful from a conservation standpoint because it may alert, for example, of an aging population with very few juveniles indicating a potential risk of decline and extinction. The age of Barbourula busuangensis from museum specimens was estimated by Roček et al. (2016) using skeletochronology. By counting the LAGs (lines of arrested growth) in mid-femoral cross sections, one individual measuring 59 mm (SVL) was considered a 4–5-year-old subadult, and another of SVL = 66 mm, a 4–6-year-old adult (Roček et al., 2016). However, the smallest gravid female in the field measured 51.2 mm (SVL), confirming that this species can reach sexual maturity around this size, similar to Bombina bombina, another frog of the family Bombinatoridae (Cogalniceanu & Miaud, 2003). Furthermore, growth rates from recaptured individuals in this study and corresponding age extrapolations indicate that a young adult of SVL = 51–55 mm would be approximately 2.5 years old (Table 2), much younger than estimations from skeletochronology by Roček et al. (2016). We found that growth rate decreased significantly with body size, suggesting that young B. busuangensis allocate more energy to growth than adults (Fig. 5), consistent with findings in other aquatic anurans (e.g., Ryser, 1988; Tsiora & Kyriakopoulou-Sklavounou, 2002; Cogalniceanu & Miaud, 2003; Erismis, 2018). Adults grew on average 62% slower than subadults (0.60 vs. 1.56 mm/month) and ranged in body size from 50 mm at sexual maturity and 2.5 years of age, up to 95.3 mm in the largest individual captured at an estimated age of approximately 11 years (Table 2). The majority of the adult population at both sites was comprised of individuals in Adult-70 and Adult-90 categories indicating that survivorship is high until these frogs reach 6–10 years of age, and decreases rapidly thereafter (Fig. 4, Table 2). The smaller bombinatorid frog Bombina bombina showed similar longevity, up to 10 years, and an age structure also dominated by mid-aged adults (Cogalniceanu & Miaud, 2003). However, because growth rate decreases with age, larger individuals could live many more years with no noticeable increase in size, and not even skeletochronology could provide accurate results since LAGs fail to accumulate in an annual basis in older individual anurans (Sinsch, 2015). Thus, an age of 11 years for Adult-90 categories is likely a conservative estimate, and the largest B. busuangensis frogs might be much older.

By defining age categories by body size (despite the limitations of this method), we were able to get an idea of the age structure of Barbourula busuangensis and infer on other aspects of the biology of the species (Fig. 4). For example, a greater number of adults than subadults might indicate that populations are declining due to low recruitment (Middleton & Green, 2015). However, the robust population size numbers obtained for both sites and over the three sampling periods suggest that this is not the case for B. busuangensis in Busuanga (Tables 3 and 4). A higher likelihood of subadults to disperse (Table 4, Appendix S6, S7 and S8), combined with high survival rates and longevity of adults may explain this outcome. In addition, Palawan still provides clean, fast flowing rivers shaded by forest canopy that provide adequate habitat for this species. This underscores the value and need to maintain the conservation practices concurrent with being an UNESCO Man and the Biosphere reserve since 1990.

Site fidelity and the ability to home has been reported for many amphibians in different lineages (see for example Crump, 1986; Gonser & Woolbright, 1995; Pittman et al., 2008; Luger, Hödl & Lötters, 2009; Ringler, Ursprung & Hödl, 2009; Matthews & Preisler, 2010; Señaris et al., 2023). In this study we recaptured 51% of the translocated Barbourula busuangensis within a 5 m radius of the original capture site, which provides experimental evidence for potential homing behavior in this species (an observation previously mentioned by Schoppe & Cervancia (2009)). We found this regardless of age, distance from original capture site, or direction of current in fast flowing rivers (Tables 4, Appendix S10). To better address this question would require knowledge of the home range of this species including migrations to, for example, reproductive sites (Sinsch, 1990). However, B. busuangensis is a fully aquatic species that reproduces within rock caves or crevices that occur in the middle of the river (Miñarro et al., 2024). Thus, frogs can find valuable breeding sites, worth homing to, within the marked transects of our study site. The fact that we recovered less individuals when they were displaced the farthest (50 m), may suggest that there is a threshold distance (around 50 m) upon which frogs are unable to home. However, farther displacements were done during our last field season (April–June 2023) rendering less opportunities to recapture these individuals. Further research is necessary to determine the maximum homing distance for this species. Other anurans adapted to torrent environments, such as Amolops mantzorum, Conraua goliath, Hylodes phyllodes, Litoria nannotis, and certain species of Odontobatrachus, also display site fidelity and, in some cases, homing behavior (see, respectively, Hodgkison & Hero, 2001; Liao, 2011; Alencar et al., 2012; Schäfer et al., 2021; Gonwouo et al., 2022), suggesting that this behavior may have an adaptive value to those anurans regardless of the difficulties of the habitat type.

In addition to contributing reliable information about animal population abundances and address other questions on species ecology and behavior, CMR methods can identify conservation risks confronted by amphibians (Donnelly & Guyer, 1994). For example, using CMR methods, Lampo et al. (2012) and Newell, Goldingay & Brooks (2013) were able to identify population increases after a decline in stream breading frogs in Venezuela and Australia respectively; Reyes-Moya, Sánchez-Montes & Martínez-Solano (2022) revealed the importance of functional connectivity to inform conservation strategies for six species of pond dwelling amphibians in Spain; and Cole et al. (2014) pointed the detrimental effects of forest disturbance on the population status of Pristimantis populations in the Andes. In this study, results from multistate models informed by CMR data suggest that populations of Barbourula busuangensis in Busuanga are enduring at the time, characterized by similar abundance estimates at north (San Rafael) and southern (Malbato) sites, and constant high survival probabilities. Furthermore, by applying these methods we discovered that this species returns to specific sites when displaced, has an age structure biased toward mid-aged adults that grow at slower rates than younger frogs, and that while subadults are more likely to migrate from the population, adults tend to stay and may live over 11 years. Our work highlights the importance of natural history and demographic studies to reveal population dynamics and unknown behaviors in poorly studied frogs, like the enigmatic, ancient frog, B. busuangensis. This kind of research contributes to improving our understanding of the status of species that are considered Data Deficient or Nearly Threatened (by IUCN), and help to determine when conservation management strategies need to be prioritized.

Supplemental Information

Supplemental Information 1 Output table of the adjusted linear model that explores the relationship between the SVL of individuals at their first capture, and the growth rate calculated upon subsequent recaptures.

Supplemental Information 2 Results of all models generated by the overall POPAN analysis in Malbato and San Rafael across the entire study period.

Models are ranked by their AICc value. Apparent survival of individuals is coded as ϕ, p is the probability of capture, and pent is the rate of entrance of new individuals into the study area. A period (.) is used to represent the parameters that are kept constant in the model, and a (t) for parameters that were modelled as time-dependent.

Supplemental Information 3 Estimates for the parameters ϕ, p, pent and N for all models with AICc weight > 0.05 from Malbato over our study period.

Apparent survival of individuals is given by ϕ, p is the probability of capture, pent the rate of entrance of new individuals in the study area between two sampling occasions and N is the estimated abundance of B. busuangensis.

Supplemental Information 4 Estimates for the parameters ϕ, p, pent and N for all models with AICc weigh > 0.05 from San Rafael over our study period.

Apparent survival of individuals is given by ϕ, p is the probability of capture, pent the rate of entrance of new individuals in the study area between two sampling occasions and N is the estimated abundance of B. busuangensis.

Supplemental Information 5 Results of all models generated by the overall POPAN formulation, ranked by the AICc value, of B. busuangensis in Malbato during the three sampling seasons (April–June 2022, October–November 2022 and April–June 2023).

Apparent survival of individuals is given by ϕ, p is the probability of capture, and pent the rate of entrance of new individuals in the study area between two sampling occasions. A period (.) is used to represent the parameters when kept constant, (t) for parameters that were modelled as time-dependent, (g) for those dependent on age class and (g*t) represents their interaction.

Supplemental Information 6 All estimates for the parameters ϕ, p, pent and N for all models with AICc weigh > 0.05 from the site of Malbato during the first sampling season (April–July 2022).

Apparent survival of individuals is given by ϕ, p is the probability of capture, pent the rate of entrance of new individuals in the study area between two sampling occasions and N is the estimated abundance. For all parameters “1” designates subadults and “2” adults.

Supplemental Information 7 All estimates for the parameters ϕ, p, pent and N for all models with AICc weigh > 0.05 from the site of Malbato during the second sampling season (October–December 2022).

Apparent survival of individuals is given by ϕ, p is the probability of capture, pent the rate of entrance of new individuals in the study area between two sampling occasions and N is the estimated abundance. For all parameters “1” designates subadults and “2” adults.

Supplemental Information 8 All estimates for the parameters ϕ, p, pent and N for all models with AICc weigh > 0.05 from the site of Malbato during the third sampling season (April–June 2023).

Apparent survival of individuals is given by ϕ, p is the probability of capture, pent the rate of entrance of new individuals in the study area between two sampling occasions and N is the estimated abundance. For all parameters “1” designates subadults and “2” adults.

Supplemental Information 9 Standardized log-odds-ratio (LOR) chi square (X2) statistics and the associated two-sided-p-values to test for “transience” and “trap-dependence” for MARK formulation at both sampling sites and during each sampling period at Malbato.

Significant results are marked in bold.

Supplemental Information 10 Output table of the multiple logistic regression model, with age (subadult vs adult), distance (10, 30 or 50 m) and direction (upstream or downstream) as predictors and homing success as the dependent variable.

Supplemental Information 11 Raw data: all marked and recaptured Barbourula busuangensis over all studied periods.

Date of collection and subsequent marking (y/m/d); genus and species name; transponder ID; recap (y/n); VIE color codes; time of collection; snout-to-vent length (SVL, mm); age (juvenile, subadult or adult); weight (g); sex (unknown/female); reproductive notes (gravid/not); transect where the individual was found; substrate where the individual was found; season (dry, transition or wet); and, weather of sampling day.

Supplemental Information 12 Raw Data. The different matrix (recapture history) for both sites (Malbato and San Rafael) used in the program MARK.

Sheet1: matrix generated for Malbato; sheet2: input matrix for MARK; sheet3: matrix generated for San Rafael; and, sheet4: input matrix for MARK. 1= marked/recaptured, 0= no recapture.

We are thankful to S. Castroviejo-Fisher for his constructive comments on an earlier version of this manuscript. We also thank A. Sánchez, G. Mochales, H. Martínez, Í. Ruiz, J. Aznar, J. Goyes, M. Gutiérrez, M. Guilbaud, R. Cueva, T. Aledón and V. Marques for their help in the field when they visited MM in Busuanga. Additionally, we thank the two anonymous reviewers for their comments which substantially improved this contribution. We are especially grateful to Kuya Lito and Kuya Archie, our regular field assistants at Malbato and San Rafael respectively; without them our recapture rate would have been much lower. Finally, we must acknowledge Manny Reyes and Tunggay Reyes (King Fisher Park) for facilitating our living quarters and boarding in Malbato.

Additional Information and Declarations

Competing Interests

Author Contributions

Animal Ethics

Data Availability

The authors declare that they have no competing interests.

Marta Miñarro conceived and designed the experiments, performed the experiments, analyzed the data, prepared figures and/or tables, authored or reviewed drafts of the article, and approved the final draft.

Patricia Burrowes conceived and designed the experiments, analyzed the data, prepared figures and/or tables, authored or reviewed drafts of the article, and approved the final draft.

Claudia Lansac analyzed the data, prepared figures and/or tables, authored or reviewed drafts of the article, and approved the final draft.

Gregorio Sánchez-Montes analyzed the data, prepared figures and/or tables, authored or reviewed drafts of the article, and approved the final draft.

Leticia E Afuang conceived and designed the experiments, authored or reviewed drafts of the article, contributed to the complex task of obtaining national and regional permits to conduct our research in the Phillippines, and approved the final draft.

Ignacio De la Riva conceived and designed the experiments, authored or reviewed drafts of the article, and approved the final draft.

The following information was supplied relating to ethical approvals (i.e., approving body and any reference numbers):

Palawan Council for Sustainable Development (PCSD, Republic of the Philippines), provided full approval for this research (Wildlife Gratuitous Permit No. 2022-11, Certification No. 2023-04).

The following information was supplied regarding data availability:

The raw data are available in the Supplemental Files.

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
