# Peer review of "Demography and homing behavior in the poorly-known Philippine flat-headed frog Barbourula busuangensis (Anura: Bombinatoridae)"

_PeerJ, doi:10.7717/peerj.18694_

## Round 0.1 · original submission · Major Revisions

Thank you very much for your manuscript titled “Demography and homing behavior in the poorly-known Philippine flat-headed frog Barbourula busuangensis (Anura: Bombinatoridae)” that you sent to PeerJ.

This study presents very valuable and relevant demographic information on the flat-headed frog, Barbourula busuangensis in two locations from the Philippines.

As you will see below, comments from referee 1 suggest a major revision while reviewers 2 suggests a minor revision before your paper can be published. Given this, I would like to see a major revision dealing with the comments. Their comments should provide a clear idea for you to review, hopefully improving the clarity and rigor of the presentation of your work. I will be happy to accept your article pending further revisions, detailed by the referees.

Reviewer 1 suggests clarifying the hypotheses and correcting some statistical analyses and limiting them to the results found. In addition, the discussion should consider the environmental pressures that can shape the natural history of the species studied. This reviewer also considers modifying some of the tables and figures.

Reviewer 2 has some methodological doubts and recommends the use of some R packages that may be useful in future studies.

Please note that we consider these revisions to be important and your revised manuscript will likely need to be revised again.

Reviewer 1 ·

Basic reporting

In general, the text is well written and understandable, and the language is appropriate. However, it is necessary to revise the English to use more direct phrases and fewer superlatives.
The article includes a good literature review. Some ideas lack citations to support the authors' claims. It is framed within a clear conceptual framework and incorporates relevant literature in the field.
The text is well-structured, follows the journal's guidelines, and shares the raw data. Some of the tables and figures require modifications in terms of style, color tones, font type, and the selection of the color palette. The study contains very relevant and interesting information; however, some hypotheses are not clearly defined.

Experimental design

The importance of the article and its purpose are clear, but the research question and the hypotheses for each of the evaluated aspects are not clearly defined. In the section on the transponder implantation, there is no mention of the protocol followed to ensure the well-being of the individuals.The methodology, in general, aligns with the purpose of the article. However, there are methodological flaws concerning the sampling design, with one site being sampled much more than another. It is unclear how the authors control for bias due to differences in sampling effort between sites. Additionally, the statistical analysis does not clearly specify the dependent and independent variables, as well as the random effects. The section on homing contains significant methodological flaws that compromise the robustness of the statistical analyses.

Validity of the findings

Certain statistical analyses need to be corrected, and the limitations of some results should be clarified, considering methodological issues and data analysis that may influence the conclusions. Overall, these adjustments would not change the findings of the study, which are very interesting and important for understanding the ecology of anurans in different parts of the world. Additionally, it is important to contextualize the species, the environmental conditions in which it lives, and to link potential evolutionary pressures that may be shaping the aspects of natural history and movement ecology observed in this article.

Annotated reviews are not available for download in order to protect the identity of reviewers who chose to remain anonymous.

Reviewer 2 ·

Basic reporting

The writing is clear an unambigious, there is sufficient background and context from the literature, the paper is well written.

Experimental design

Clear Aims and Scope, with clear relevant research questions. Robust classic CMR methods.

Validity of the findings

Clear impact and novetly for a poorly understood and charismatic frog species. Ideally the raw data matrix/ spreadsheet should be shared.

Additional comments

Dear Editors,
The manuscript titled “Demography and homing behavior in the poorly-known Philippine flat-headed frog Barbourula busuangensis (Anura: Bombinatoridae)” by Marta Minarro et al. presents a comprehensive and detailed research into the population dynamics and homing behavior of Barbourula busuangensis, a species endemic to the province of Palawan in the Philippines. Utilizing robust capture-mark-recapture methods and advanced statistical modeling, the authors have successfully provided valuable insights into the species’ demographics, survival rates, and site fidelity. The findings of this research are significant, offering a crucial baseline for future conservation efforts and enhancing our understanding of this poorly-known amphibian species, and worthy of publication.
I have some minor comments below which I hope will improve the manuscript on its journey to get published.
Some minor comments:
Did you not seal the wound with some form of skin glue? Any evidence of tag loss during the project?
Program Mark and the method used for the analysis is valid and publishable, just want to point to the authors to some R packages that can, for next time be of interest. I’m currently playing with Bayesian mark recapture, and it seems to have some added features/ power worth exploring, for example::
1. marked: This package allows for the analysis of mark-recapture data using both maximum likelihood estimation (MLE) and Bayesian Markov Chain Monte Carlo (MCMC) methods. It supports various models, including the Cormack-Jolly-Seber (CJS) model1.
2. RMark: While primarily an interface to the MARK software, RMark can be used to construct models for MARK, which includes Bayesian analysis capabilities. It allows for the creation of input files for MARK and extraction of output2.
3. BTSPAS: This package provides advanced Bayesian methods to estimate abundance and run-timing from temporally-stratified Petersen mark-recapture experiments. It includes hierarchical modeling of capture probabilities and spline smoothing of daily run size3.
4. NIMBLE: Although not exclusively for mark-recapture, NIMBLE is a flexible package for building and fitting Bayesian hierarchical models using MCMC. It can be used to implement custom Bayesian mark-recapture models4.
Is there a practical way in future to define sex? There are now cheaper portable ultrasound machines, perhaps its time we herpetologist play about more with these with species like yours.

You got some data missing perhaps? Line 231, ‘corresponding to XX adults and XX subadults’
Figure 2. The contours of the sea floor are not important and distract, can you perhaps improve the country/ region wide maps? Where is Manila to give the naïve reader some context? A graticule for the country/ region wide insets would also be welcome.

Otherwise this work is great, exactly what we need for these poorly known/ understood amphibian species.

---

## Round 0.2 · accepted · Accept

After reviewing this revised version of your manuscript,I see that each of the comments was answered in detail, while the suggestions not considered are justified in detail. Therefore, I am satisfied with the current version and consider it ready for publication.